# Investigating the Coupling Effect of High Pressure and Hot Air on External Friction Angle Based on Resistance Reduction Tests on Subsoiling Tillage Tools for Sandy Clay Loam

Kuan Qin [1], Yongzheng Zhang [1], Zhougao Shen [2], Chengmao Cao [1,*], Zhengmin Wu [2], Jun Ge [1], Liangfei Fang [1] and Haijun Bi [2]

[1] School of Engineering, Anhui Agricultural University, Hefei 230036, China
[2] State Key Laboratory of Tea Plant Biology and Utilization, Anhui Agricultural University, Hefei 230036, China
* Correspondence: caochengmao@sina.com; Tel.: +86-136-9651-5592

**Abstract:** Sandy clay loam has the characteristics of both sand and clay. Because of these characteristics, both frictional resistance and adhesive resistance occur between the soil and tillage tool. The combined effect of the two frictional forces increases the external friction angle between the soil and tillage tool, thus increasing the working resistance. To address this issue, this study investigated the coupling effect of high pressure and hot air on the external friction angle by using a self-developed device to measure the external friction angle. Test results showed that high-pressure air between the soil and tillage tool formed a high-pressure air curtain layer which acted as a lubricant, thereby reducing the external friction angle. The external friction angle decreased as the airflow pressure increased. The reduction in the moisture content of the subsoil to less than 30% by the high-pressure hot air reduced the resistance between the soil and tillage tool. The approach with the high-pressure hot air curtain was verified in tests on a subsoiling shovel; the working resistance of the shovel under high-pressure hot air was reduced by 14.8%, demonstrating that this approach was effective in reducing the working resistance of the shovel.

**Keywords:** sandy clay loam; high pressure; hot air; external friction angle; resistance reduction

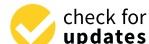



## 1. Introduction

Sandy clay loam is widely used worldwide for growing staple food crops and cash crops [1–3]. It has the characteristics of both sand and clay [4]. When using tillage tools (e.g., subsoiling shovel and plough) on sandy clay loam, both frictional resistance and adhesive resistance occur between the soil and tillage tool surfaces during operation [5–7]. The combined effect of the two frictional forces increases the external friction angle between the soil and tillage tool [8,9]. This in turn increases the coefficient of friction between the soil and tillage tool surfaces [10], thus further increasing the working resistance [11,12].

Researchers have adopted various methods to reduce the external friction angle between the soil and tillage tool surfaces during operation to reduce the working resistance. For example, inspired by the scales on the skin of sharks, Wang et al. [13] designed riblet structures on the surface of a shovel to create a gap between soil and tillage tool surfaces, in which air was stored. This stored air acted as a lubricant that reduced the adhesion of the metal's surface to the soil, thereby reducing frictional and adhesive resistance. Regarding the subsoiling shovel, Zhao et al. [14] applied self-excited vibrations that were perpendicular to the soil-surface direction during the operation of a shovel. Zhang et al. [15] used connecting rods to enable the circular reciprocating motion of a shovel during operation. The above vibration method can be used to reduce soil compaction on the surface of the subsoiling shovel, thereby reducing the contact area. In addition, the overflow of moisture and air from the soil adds lubrication between the soil and shovel, which reduces adhesion. Zhang et al. [16] have found that the mucus of earthworms passing through the

soil has some lubricating effect which can reduce soil adhesion and frictional resistance. The coating of this mucus on subsoiling tools can reduce the working resistance of the tools. Guan et al. [17] applied a new TiAlN/CrN coating to the surface of a tillage tool to improve the hydrophobicity of the tool surface, thereby reducing adhesion and resistance. Yan et al. [18] adopted an electroosmotic design to reduce adhesion and friction. Specifically, a voltage was applied to the tillage tool which forced water to travel from the soil to the interface between the soil and the shovel.

These methods reveal that the key to reducing the frictional resistance between tillage tools and soil is to form a lubricating layer of air or water between the two surfaces [19]. To reduce the adhesive resistance between the tool and soil surfaces, it is crucial to reduce the surface tension of water molecules between them and to improve the hydrophobicity of the surface of the tillage tool [20,21]. For sandy clay loam, the frictional and adhesive resistances must be addressed to reduce the external friction angle of the tillage tool. The fundamental method of reducing frictional resistance is to use air with high mobility and low friction as a lubricant between the tool and soil [22]. In contrast, the fundamental method of reducing adhesive resistance is to dry the soil in contact with the tillage tool using heat, thereby reducing the moisture content of the soil and the surface tension of the water in the soil [23]. Given the aforementioned considerations, the gap between the tillage parts and soil may be filled with high-pressure hot air to form a high-pressure hot air curtain, which has lubricating and drying effects, thereby reducing both frictional and adhesive resistance. This is a new way of reducing the resistance, which can effectively reduce the external friction angle and working resistance between sandy clay and tillage tools. However, there is no quantitative experimental study on sandy clay loam based on this method at present.

In this study, experiments were conducted to investigate the causes and extent of reduction in the external friction angle when high-pressure hot air is filled between the tillage tool and soil. The external friction angle between the soil and tillage tool is usually measured using an inclined plane test [24]. Specifically, the soil is placed on an inclined surface at a certain angle; the angle of inclination of the surface when the soil is in the critical state of sliding is taken as the angle of external friction [25,26]. In this study, equipment that can produce high-pressure hot airflow was developed to measure the external friction angle and to determine the change in the external friction angle between the soil tillage tool surfaces under various pressure temperatures, thereby investigating the effect of the high-pressure hot air curtain on the external friction angle and resistance reduction using a subsoiling shovel. According to existing research, the area of contaminated soil has increased due to the misuse of pesticides and other chemicals. The application of hot air to the soil can promote the volatilization of organic matter and increase the infiltration rate of the soil, which can help restore the contaminated soil [27]. The results of this study can provide both theoretical and experimental references for the application of the high-pressure hot air curtain in resistance reduction for tillage tools.

## 2. Materials and Methods

### 2.1. The Setup for the Measurement of the External Friction Angle

Figure 1 shows the equipment used for measuring the external friction angle; namely, an air pump, a hot blast stove, and a device developed to measure the external friction angle. The high-pressure air from the air pump flows to the hot blast stove through a vent pipe (made of PU, resistant to 120 °C, 800 mm in length, 6 mm in diameter), where it is heated to a high temperature and sent to the device developed to measure the external friction angle.

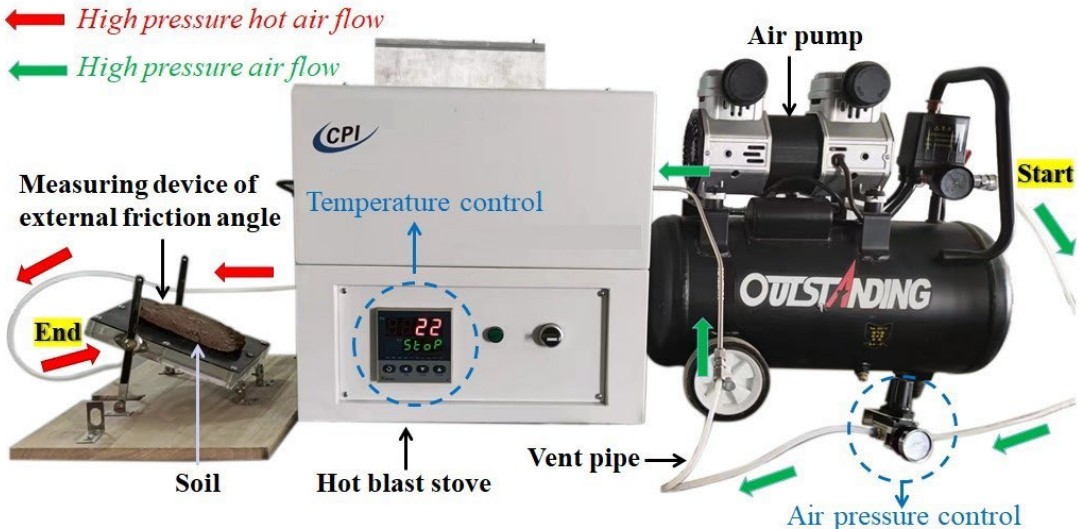

**Figure 1.** Experimental setup for measurement of external friction angle.

Parameters of the air pump and hot blast stove are listed in Table 1. An air pressure valve is fitted to the vent pipe of the air pump to control the air pressure from the air tank to the vent pipe. A pressure gauge is equipped at the entrance of the device used to measure the external friction angle for reading the input pressure. The high-pressure air produced from the air pump is pumped into a high-temperature-resistant metal vent pipe inside the hot blast stove, where the air is heated to a certain temperature inside the metal pipe and then passed to the measuring device through the vent pipe.

**Table 1.** Parameters of the air pump and hot blast stove.

| Air Pump | | Hot Blast Stove | |
|---|---|---|---|
| **Parameters** | **Values** | **Parameter** | **Value** |
| Rated power (W) | 1600 | Heating power (W) | 6000 |
| Exhaust pressure (MPa) | 0.05–0.85 | Temperature (°C) | 0–620 |
| Storage capacity (L) | 30 | Temperature control accuracy (°C) | 1 |

Views of the self-developed device for measuring the external friction angle are shown in Figure 2. The main body of the measuring device is a cuboid with an inner cavity. The device is 200 mm in length, 150 mm in width, and 20 mm in height. The upper surface plate of the device is made of 65 Mn steel, which is the material typically used for tillage tools. On the plate, $15 \times 9$ stomata with a diameter of 1 mm are evenly arranged, with a spacing of 10 mm between each two stomata. The inner cavity of the hollow cuboid is enclosed by transparent acrylic panels, which form the sides and bottom of the cuboid. Every part of the device is sealed, except for the stomata. The bottom surface of the cuboid is connected to a vent pipe that feeds the high-pressure hot air from the hot blast stove into the inner cavity. This air eventually exits from the upper surface of the cuboid. The entire cuboid is mounted on a flat base with the low end fixed. The sides of the cuboid are attached to slide rails and the cuboid can be slid along the rails to adjust the angle φ (0° to 80°) between the cuboid and base. This angle is defined as the angle of external friction between the soil and tillage tool in the measurement [28].

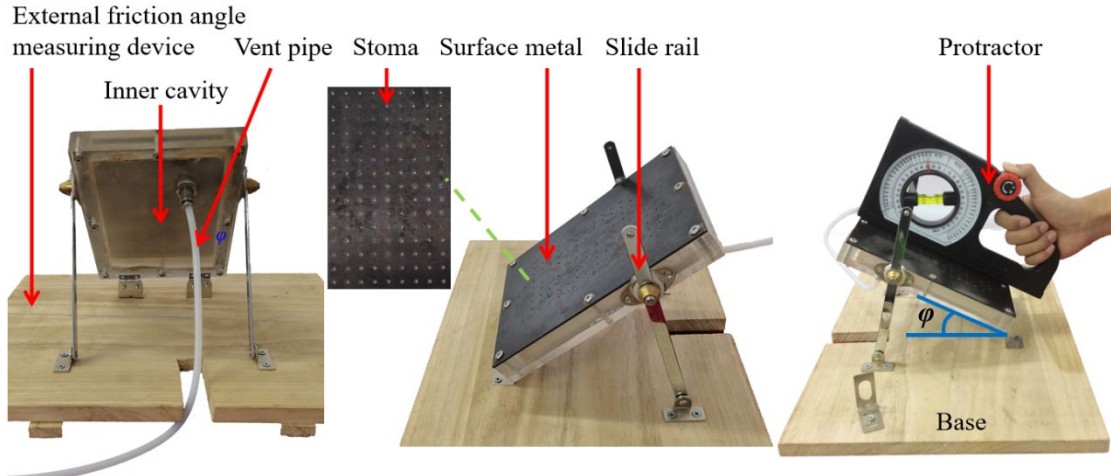

**Figure 2.** Views of the self-developed device for measuring external friction angle.

*2.2. Soil Parameters and Sample Preparation*

To investigate the variations in the external friction angle under the effect of a high-pressure hot air curtain [29], soil samples with different moisture contents were prepared (at 15%, 20%, 25%, 30%, 35%, and 40%). The soil samples were obtained from the experimental field in the Agricultural Park of Anhui Agricultural University in Hefei City, Anhui Province, China. The soil moisture content was measured at 25%. The soil was a sandy clay loam; its parameters are shown in Table 2 (classification according to international soil classification standards). The soil used to make the samples was weighed to ensure that each soil sample was of the same quality. To fit soil samples into the rectangular surface of the device for measuring the external friction angle, soil samples with dimensions of 150 mm × 100 mm × 20 mm (L × W × H) were prepared using a cuboid mold, thereby maintaining consistency in dimensions. Soil samples with different moisture contents were obtained by adding different amounts of water or allowing water evaporation in a natural state. The moisture content of soil was measured using the IJ-100F moisture meter, which has a precision of 0.1%. During the test, the moisture content of soil samples were measured regularly with the moisture meter, the samples sprayed with water regularly to ensure the stability of their moisture content.

**Table 2.** Parameters of the soil sample.

| Soil Parameters | Values |
|---|---|
| Moisture content | 25.0% |
| Soil compaction | 1516 kPa |
| Soil bulk density | 1920 kg/m$^3$ |
| Soil content (Clay, silt, and sand contents) | 38.1%, 25.7%, 36.2% |
| Cohesion | 15.67 kPa |

*2.3. Methods*

This study investigated the coupling effect of high pressure and hot air on the external friction angle by using a self-developed device to measure the external friction angle. Finally, the approach with the high-pressure hot air curtain was verified in tests on a subsoiling shovel. Before the formal test, the pre-test was conducted several times and the average value of several tests was sought in the formal test to reduce the error.

### 2.3.1. Tests for Air Pressure on the External Friction Angle

Experiments were conducted to investigate the effect of airflow pressure on the external friction angle by varying pressure of the air filling the space between the soil and tillage tool. The pressure of the output airflow was controlled by the air pressure valve of the air pump and the reading on the pressure gauge was used. Four experimental groups and a control group were used for the tests. For the experimental groups, the pressure was set to 0.1 MPa, 0.2 MPa, 0.3 MPa, and 0.4 MPa, whereas the control group had no airflow. Soil samples with varying moisture contents (15%, 20%, 25%, 30%, 35%, and 40%) were placed on the metal surface of the device used to measure the external friction angle. Airflow was pumped at different pressures (0.1 MPa, 0.2 MPa, 0.3 MPa, 0.4 MPa, and no airflow) through the stomata to fill the space between the soil sample and metal surfaces. As shown in Figure 3, the position of the cuboid of the measuring device was adjusted using the slide rails until the soil reached the critical state of sliding, at which point the external friction angle was measured. Each test involved the use of a soil sample with a specific water content and airflow at a specific pressure. Each test was repeated four times, the mean value of the test results being taken as the final result.

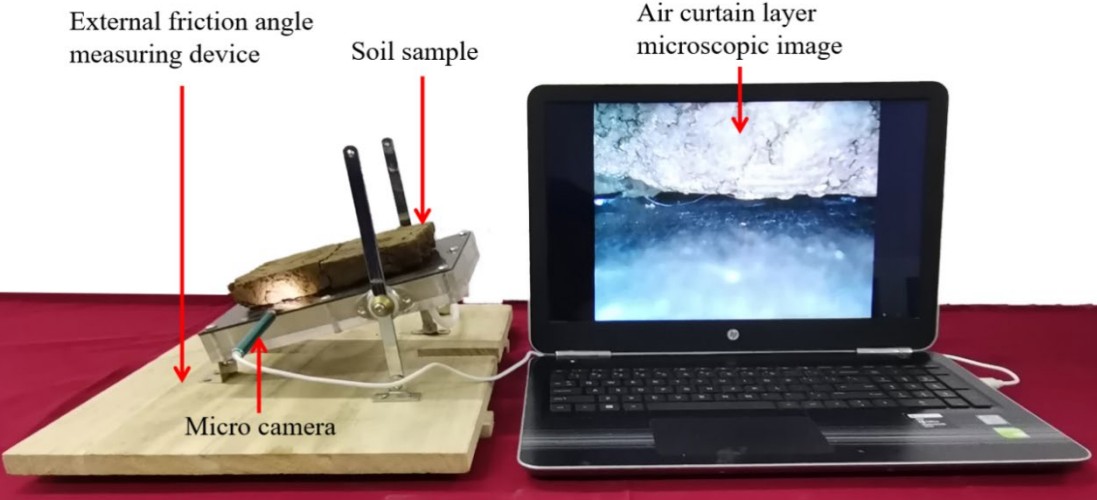

**Figure 3.** Experimental process captured by a microscopic camera.

Throughout the experiment, the eheV2 multi-purpose microscopic camera (with $20\times$ magnification) was used to record changes in the air curtain layer between the soil surface and metal surface of the device used to measure the external friction angle. Through measurements, we obtained the area and width of the air curtain layer as well as the surface roughness of the subsoil in contact with the air curtain layer [30], hereafter referred to as soil surface roughness and calculated using Equation (1). The soil surface roughness was used to analyze the factors that influenced the variations in the external friction angle between the soil and tillage tool surfaces when the air curtain was generated at varying pressures.

$$S = \sqrt{\frac{1}{n}\sum_{i=1}^{n}(a_i - a)^2} \tag{1}$$

where, $S$ refers to the soil surface roughness, represented as the standard deviation; $a_i$ denotes the distance from the subsoil surface at the measuring point to the metal surface of the device used to measure the external friction angle; $a$ denotes the average distance from the subsoil surface to the metal surface of the device used to measure the external friction angle at all measuring points; and $n$ is the number of measuring points. In this experiment, 10 measuring points were marked on the surface of the subsoil.

### 2.3.2. Testing the Coupling Effect of High Pressure and Hot Air on the External Friction Angle

Based on the previous tests for the effect of air pressure on the external friction angle, the study further investigated the coupling effect of high pressure and airflow at different temperatures on the external friction angle using the same method. In these tests, the air pressure was set to the value that had resulted in the greatest reduction in the external friction angle. Soil samples were placed on the metal surface of the device used for measuring the external friction angle. After the air pressure was stabilized, the temperature of the high-pressure hot air released from the stomata was set to 40, 60, 80, and 100 °C for different samples by adjusting the temperature of the hot blast stove. The temperature of the hot air was measured using the Testo835-T1 infrared thermometer, with an accuracy of ±1 °C. In the experiment, temperature was the variable; the four temperature values above were taken for the experimental groups with a heating time of 5 s. For the control group, no heating was applied. Soil samples with different moisture contents (i.e., 15%, 20%, 25%, 30%, 35%, and 40%) were prepared.

An FOTRIC-220S thermal imager (with a temperature range of −20 °C to 650 °C and an accuracy of 2 °C) was used during the entire experiment to obtain thermal images when the soil was heated and dried by the high-pressure hot air generated from the metal surface of the device used to measure the external friction angle, as shown in Figure 4. After the experiment, we obtained the subsoil temperature and temperature gradient of the high-pressure hot air curtain (hereinafter referred to as the temperature gradient), which indicates the change in temperature along the direction perpendicular to the metal surface. In addition, an IJ-100F moisture meter (with an accuracy of 0.1%) was used to measure the moisture content of the soil after heating. Hot air acted directly on the subsoil that was in contact with the metal surface; therefore, the moisture content of the subsoil surface that was in contact with the metal surface was taken as the moisture content of the subsoil at that measuring point, hereafter referred to as subsoil moisture content. This parameter can be used to analyze the causes of changes in the external friction angle after the high-pressure hot air at different temperatures filled the space between the soil and tillage tool surfaces.

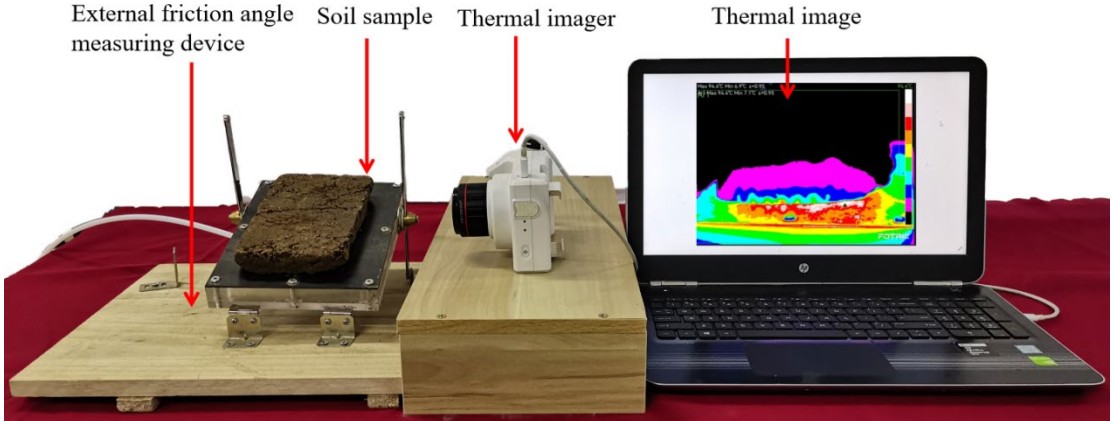

**Figure 4.** Experimental process captured by the thermal imager.

### 2.3.3. Subsoiling Test on the Coupling Effect of High Pressure and Hot Air

The subsoiling shovel is a commonly used tillage tool. The coupling effect of high pressure and hot air on the resistance reduction for the subsoiling shovel was investigated in a self-developed soil bin [31]. The setup of the soil-bin test is shown in Figure 5 and the main parameters are listed in Table 3. The test involved a mobile bench, electric machinery, a hot blast stove, an air pump, a three-way force sensor, and an electrical control cabinet. The subsoiling shovel used met ISO 5680:1979 standards. It was installed under the three-way force sensor, which was mounted under the mobile bench. Driven by the electric machinery and slide rails, the bench could move forward and backward as a whole and it

could lift or lower the shovel [32,33]. The motion parameters of the shovel were controlled by the electrical control cabinet. During the operation of the shovel, the air pump and hot air stove generated high-pressure hot air, which was applied to the shovel. The shovel was also made of 65 Mn steel, had 15 stomata with a diameter of 1 mm on its surface for the exit of high-pressure hot air, and the inside of tip of the shovel was hollow.

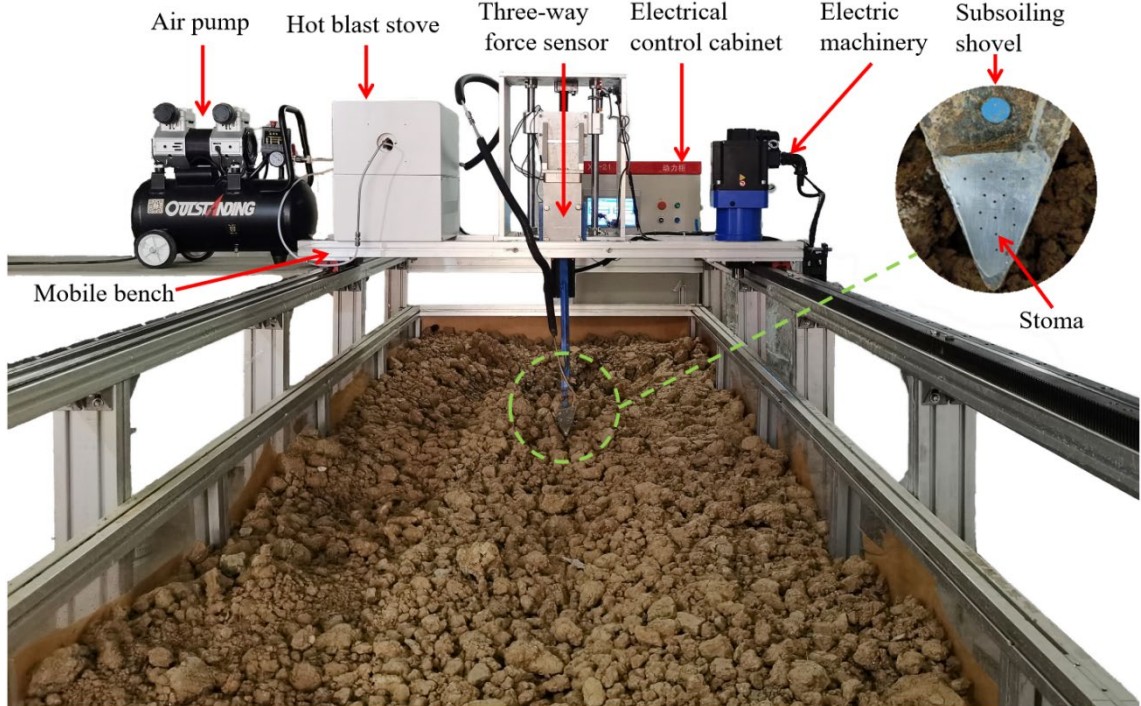

**Figure 5.** Setting of the soil-bin test.

**Table 3.** Main parameters for the soil-bin test.

| Parameters | Values |
|---|---|
| Size of the unit (L × W × H; m) | 10 × 1.2 × 1.6 |
| Operating speed (m/s) | 0–1.5 |
| Subsoiling depth (mm) | 0–300 |
| Highest air temperature at shovel surface (°C) | 100 |
| Air pressure at shovel surface (MPa) | 0–0.4 |
| Measuring range of the three-way force (kN) | $F_X = F_Y = F_Z = 5$ |

The parameters of the soil used in the soil bin test were the same as those used in the measurement of the external friction angle [34]. The moisture content of the soil was 25% (consistent with the tea garden experimental field), the shovel depth was 150 mm, the advancing speed was set to 0.2 m/s and 10 m of the soil channel was tested. The air pressure and temperature corresponding to the greatest reduction in the external friction angle in the measurement test were used for the subsoiling test under high-pressure and hot air conditions. In addition, two control groups were designed: one group with high-pressure airflow and no heating and another group with no heating and no pressure. In the test, a three-way force sensor was used to measure the resistance of the subsoiling shovel during operation [35,36]. The data were collected from the electrical control cabinet. For each group, the test was repeated three times, and the mean value of the test results was used. To ensure consistent soil conditions for each test, the soil in the channel was leveled using a microtiller and compacted using a compaction machine after each test until the soil compaction error was within 3%.

## 3. Results and Discussion

### 3.1. The Effect of Air Pressure on the External Friction Angle

It can be seen from Figure 6 that the external friction angle decreased as air pressure increased for soil samples at different moisture contents in the experimental groups. When the air pressure was 0.4 MPa, the external friction angle was the smallest for all soil samples. Specifically, the angles of external friction corresponding to soils with moisture contents of 15%, 20%, 25%, 30%, 35%, and 40% were 12°, 19°, 22°, 23°, 17°, and 12°, respectively. Compared with the control group, which did not have airflow applied, the external friction angle decreased by 17°, 12°, 13°, 22°, 24°, and 18°, respectively, for the soil samples mentioned above. These results suggest that filling high-pressure airflow between the tillage tool and soil surfaces can reduce the external friction angle. According to the results in the studies by Sindagi et al. [37] and Yu et al. [38], high-pressure airflow filled between the hull and water acted as a lubricant, which reduced the hull resistance; the higher the air pressure, the better the resistance-reduction effect. The results of this study are in line with the findings of the studies mentioned above: high-pressure airflow acted as an air curtain lubricant and had a good effect in reducing friction when filled between two objects with relative motion.

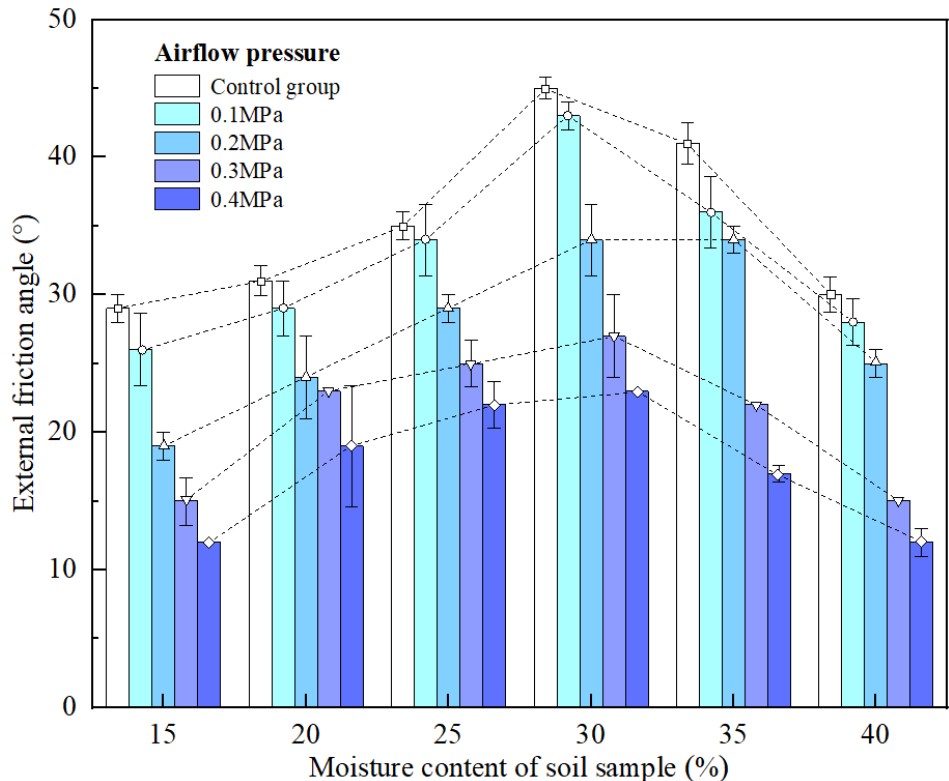

**Figure 6.** Changes in the external friction angle against airflow at varying pressures.

Irrespective of the introduction of high-pressure airflow, the external friction angle between the soil and tillage tool first increased and then decreased as the moisture content of the soil increased. The external friction angle was the largest when the soil moisture content was 30%. The greatest reduction in the external friction angle between the soil and tillage component was observed (i.e., 48.9%) when the soil moisture content was 30% and the airflow pressure was 0.4 MPa. Existing literature on the external friction angle has found that the external friction angle decreases with an increase in the moisture content of soil [39,40]. The results of this study further support these finding as a similar trend was observed after the space between the soil and tillage tool was filled with high-pressure airflow.

### 3.2. Influencing Mechanism of High-Pressure Airflow on the External Friction Angle

To further explore the mechanism behind the reduction in the external friction angle caused by high-pressure airflow, soil samples with a moisture content of 30% were used for testing. In this study, microscopic images were used to observe the air curtain layer formed between the soil and tillage tool under the effect of airflow at different pressures. Matlab was used for removing background, increasing contrast, image typology, and median filtering. The width and area of the air curtain layer (front view of air curtain layer), as well as the soil surface roughness, were analyzed to understand the mechanism by which the external friction angle was affected.

Figure 7 shows the microscopic images of the air curtain layer and the converted images after binarization processing. The parameters of the converted images were extracted. Figure 8 presents the effects of the width and area of the air curtain layer, as well as the soil surface roughness, on the external friction angle.

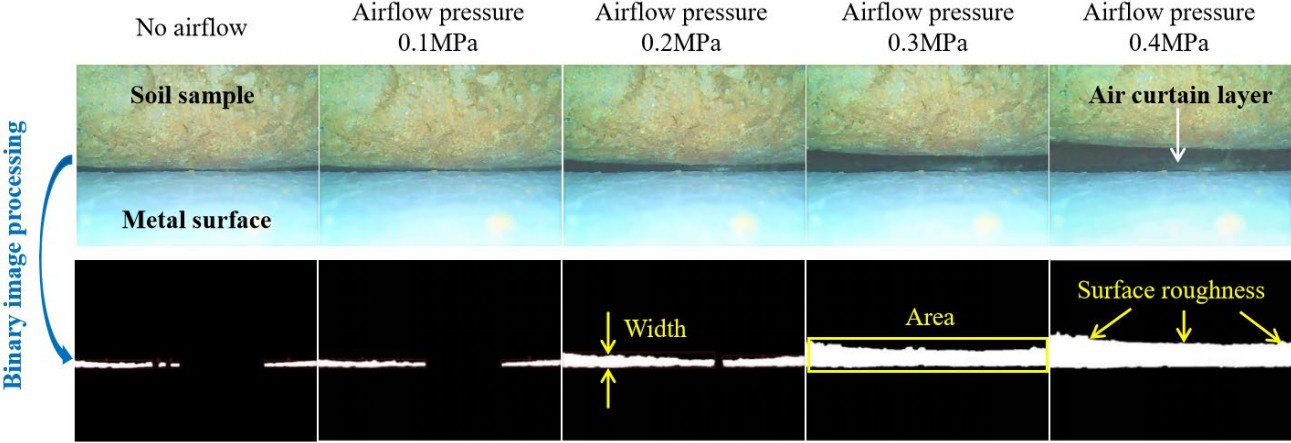

**Figure 7.** Microscopic images of the air curtain layer at different airflow pressures.

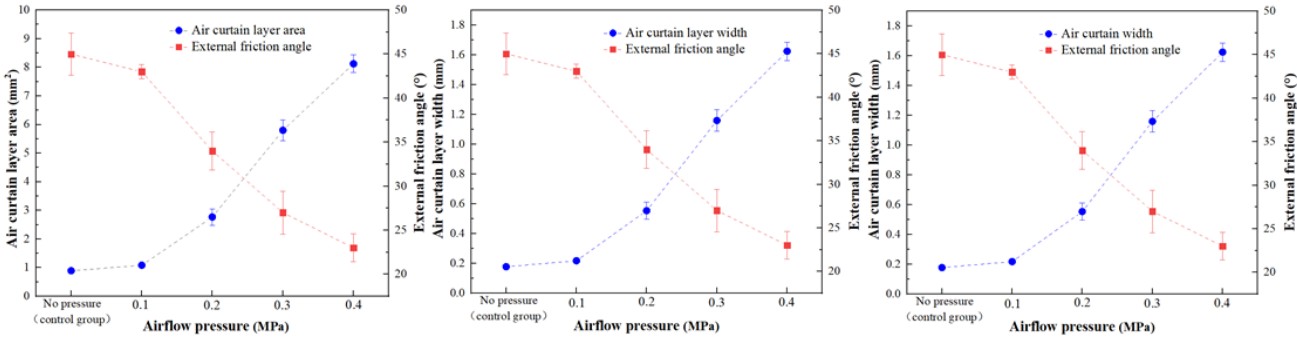

**Figure 8.** The external friction angle against the soil-surface roughness as well as the width and area of the air curtain layer.

It can be seen from Figure 8 that as airflow pressure increased, the area of the air curtain layer, the width of the air curtain layer, and soil surface roughness increased accordingly and were negatively correlated with the external friction angle. This result suggests that after the high-pressure air fills the gap between the soil and tillage tool, the external friction angle can be reduced by increasing the area and width of the air curtain layer and the roughness of the soil surface. In similar studies on the effect of high-pressure airflow on the reduction of the resistance on ships with air curtains, it was found that the surface friction between the hull and water was affected by the area of the corresponding air curtain [41], the width of the air curtain [42], and the degree of fluctuation of the liquid surface in contact with the air curtain layer [43], indicating that the above three parameters are the key factors influencing the effect of high-pressure air curtain on friction reduction.

A correlation analysis was performed on the area of the air curtain layer, the width of the air curtain layer, soil surface roughness, and external friction angle. The results of the analysis are shown in Table 4. The air curtain area and external friction angle had the highest correlation coefficient of 0.9726 while the soil surface roughness and external friction angle had the lowest correlation of 0.9407. This result indicates that the area of the air curtain layer had the greatest influence on the external friction angle, while the soil surface roughness had the least influence.

**Table 4.** Correlation analysis results of area and width of the air curtain layer, soil surface roughness, and external friction angle.

| Data Group | Air Curtain Layer Area and External Friction Angle | Air Curtain Layer Width and External Friction Angle | Soil Surface Roughness and External Friction Angle |
|---|---|---|---|
| Correlation coefficient ($r_1$) | −0.9726 | −0.9721 | −0.9407 |

According to the microscopic images of the air curtain layer in Figure 7, the width of the air curtain layer between the metal surface and soil increased significantly as the airflow pressure increased. When there was no airflow, the width of the air curtain layer was 0.178 mm. When the airflow pressure was 0.4 MPa, the width of the air curtain was 1.625 mm. Although the width was increased by only 1.447 mm, the external friction angle was reduced by 22°. This indicates that the air curtain layer formed by the high-pressure airflow between the soil and the surface of the tillage tool acted as a lubricant and had a significant effect in reducing resistance. In addition, within the pressure range allowed by the pressurizing device, the greater the airflow pressure, the wider the air curtain layer and the greater the effect of reducing the external friction angle. It can also be seen from the microscopic images that when the air curtain between the soil and metal surface became wider, the area of the air curtain layer also increased. As the airflow pressure increased, the soil surface was disturbed by the high-pressure airflow and the soil surface roughness increased. However, the external friction angle was still reduced due to the air curtain layer.

*3.3. The Effect of High Pressure and Hot Air on the External Friction Angle*

The greatest degree of reduction in the external friction angle was observed at 0.4 MPa. Under this pressure, the effect of the airflow at different heating temperatures on the external friction angle for soil samples with different moisture contents is shown in Figure 9. It can be seen from Figure 9 that as the soil moisture content increased, the external friction angle first increased and then decreased. When there was no heating, and when the heating temperature was 40 °C or 60 °C, the external friction angle reached the maximum when the soil water content was 30%. At the heating temperature of 80 °C or 100 °C, the external friction angle was the largest when the soil water content was 35%. This indicates that the external friction angle still followed the previously observed trend under heating of the high-pressure airflow. In other words, the external friction angle increased and then decreased as the soil water content increased.

For soil samples with moisture contents of 15%, 20%, 25%, and 30%, the external friction angle decreased with increasing temperature. When the heating temperature was 100 °C, the external friction angle of the soil samples mentioned above decreased by 33.3%, 21.1%, 18.2%, and 8.7%, respectively, compared to that of the control group without heating. For soil samples with moisture contents of 35% and 40%, the external friction angle increased with increasing temperature. When the heating temperature was 100 °C, the external friction angle of the soil samples mentioned above increased by 47.1% and 50.0%, respectively, compared to that of the control group without heating. Therefore, it can be inferred that 30% moisture content is the cut-off level for sandy clay loam and the heated airflow can reduce the external friction angle for soils with a moisture content of less than 30%. However, for soils with a moisture content of more than 30%, the heated airflow will increase the external friction angle.

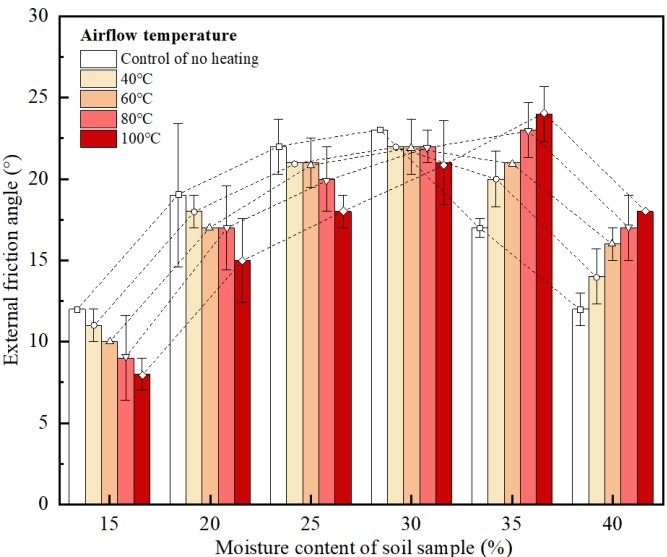

**Figure 9.** The effect of high-pressure airflow at different temperatures on the external friction angle of soils with different moisture contents.

*3.4. Influencing Mechanism of High-Pressure Hot Airflow on the External Friction Angle*

To further study the mechanism of the reduction in the external friction angle by the high-pressure hot airflow, soil samples with moisture contents of 20%, 30%, and 40% were used. The images of the hot air curtain formed by the high-pressure airflow at different temperatures and the variation in the moisture content of the subsoil were analyzed to understand the influencing mechanism of the subsoil temperature, temperature gradient, and moisture content of the subsoil on the external friction angle.

The thermal images of the soil samples are shown in Figure 10 and the effects of subsoil temperature, temperature gradient, and subsoil moisture content on the angle of external friction are shown in Figures 11–13, respectively. The schematic diagram of variations in the moisture content of the subsoils with initial moisture contents of 20% and 30% and heated at 100 °C are shown in Figure 14.

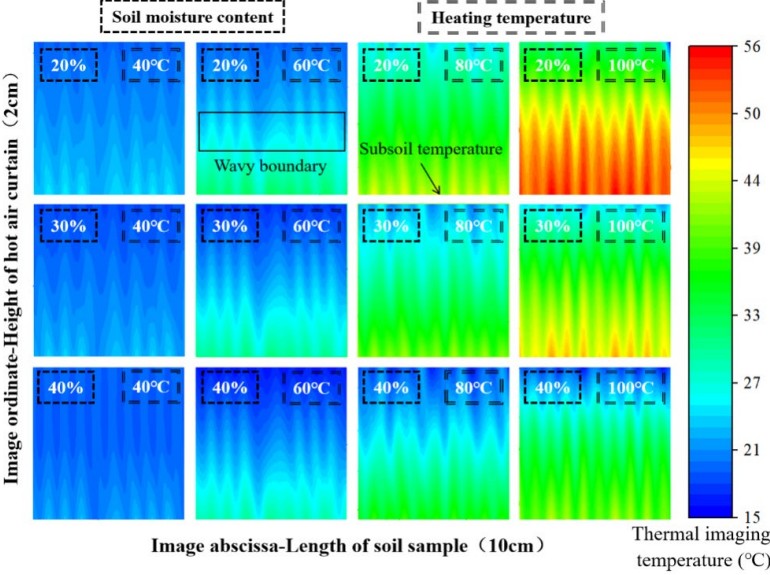

**Figure 10.** Thermal images of high-pressure hot air curtain for soil samples with moisture contents of 20%, 30%, and 40%.

As can be seen from Figure 10, the high-pressure hot airflow emitted from the stomata produced a hot air curtain layer with a temperature that decreased; the temperature at

the bottom of the air curtain layer was significantly higher than that at the top. The high-pressure hot air was pumped through isometric stomata; thus, the isothermal boundary of the decreasing temperature was wavy. In addition, as the moisture content and heating temperature varied, there was a clear changing pattern in the subsoil temperature and temperature gradient.

Figure 11 shows that for soil samples with moisture contents of 20%, 30%, and 40%, the subsoil temperature increased with increasing airflow temperature. Under the same airflow temperature conditions, the higher the soil water content, the lower the subsoil temperature. For soil samples with a moisture content of 20% or 30%, the external friction angle decreased as the subsoil temperature increased; for soil samples with a moisture content of 40%, the external friction angle increased as the subsoil temperature increased.

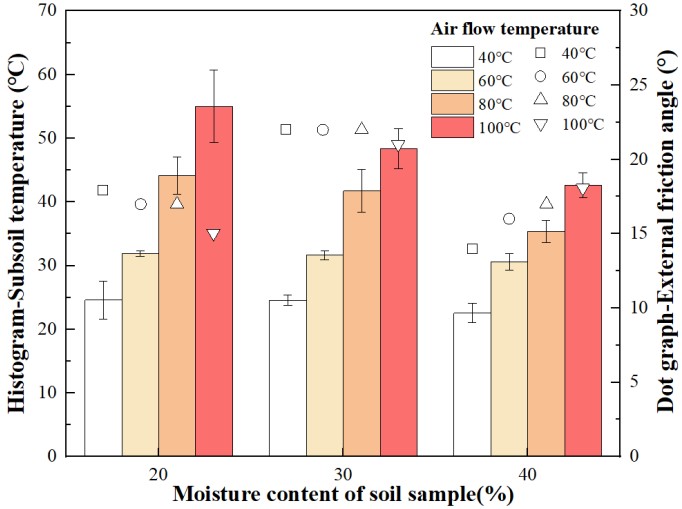

**Figure 11.** Effect of subsoil temperature on the external friction angle.

As shown in Figure 12, for soil samples with moisture contents of 20%, 30%, and 40%, the temperature gradient of the high-pressure hot air curtain increased as the airflow temperature increased. Under the same airflow temperature, no positive or negative correlation was found between soil moisture content and temperature gradient. For soil samples with a moisture content of 20% or 30%, the external friction angle decreased as the temperature gradient increased; for soil samples with a moisture content of 40%, the external friction angle increased as the temperature gradient increased.

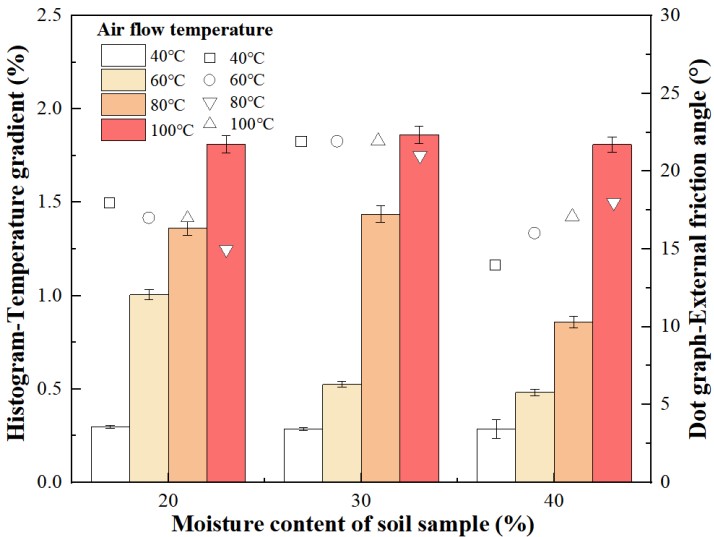

**Figure 12.** Effect of temperature gradient on the external friction angle.

Figure 13 shows that for soil samples with moisture contents of 20%, 30%, and 40%, the temperature of the subsoil decreased after heating as the airflow temperature increased. For soil samples with a moisture content of 20%, the average reduction in the moisture content of the subsoil was 9.88% below the four airflow temperature conditions. For soil samples with moisture contents of 30% and 40%, the reduction in the moisture contents was 8.75% and 5.81%, respectively. For soil samples with water contents of 20% and 30%, the external friction angle decreased as the subsoil water content decreased after heating. For soil samples with a water content of 40%, the external friction angle increased as the subsoil moisture content decreased after heating.

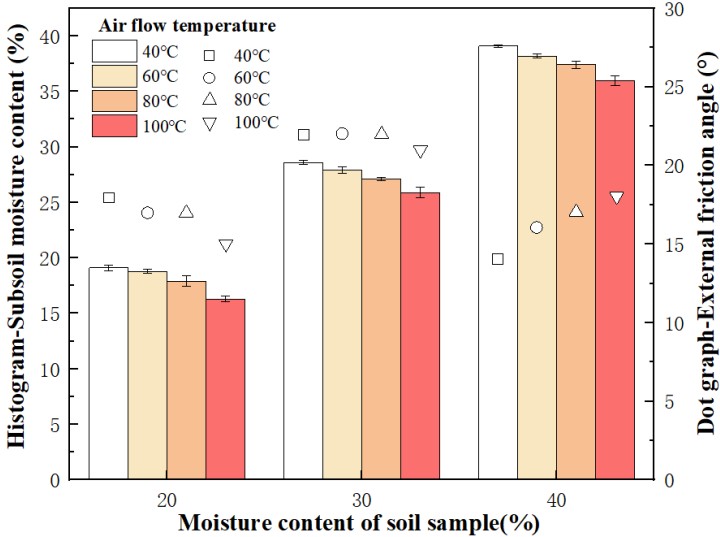

**Figure 13.** Effect of subsoil moisture content on the external friction angle.

Figure 14 shows the changes in the surface of the heated subsoils with moisture contents of 20% and 30%. It can be seen from Figure 14 that as the heating time increased, the surface color of the subsoil became noticeably lighter over time, indicating that the subsoil was being dried by the heat, and the moisture content gradually decreased. During heating, if a dry spot appeared first at a particular stoma, it was inferred that the moisture content at the stoma decreased at a faster rate. The overall moisture content of the entire soil surface decreased under the effect of the hot air curtain layer. After heating for 12 s, the subsoil was dried uniformly throughout, indicating that the high-pressure hot air had an effect on the entire subsoil. According to RGB pictures in Figure 14, drying points (3 s) and areas (9 s) occurred in a shorter period of time for soil samples with a moisture content of 20% compared to those for the soil samples with a moisture of 30%. This further explains that the soil with a lower moisture content dries at a faster speed when heated, leading to an increased reduction in the moisture content of the subsoil.

Table 5 shows the results of the correlation analysis on subsoil temperature, temperature gradient, subsoil moisture content, and external friction angle. The mean value of the correlation between subsoil temperature and external friction angle was the largest, whereas the temperature gradient had the least correlation with the eternal friction angle. This result indicates that the subsoil temperature had the strongest effect on the external friction angle, while the temperature gradient had the least impact.

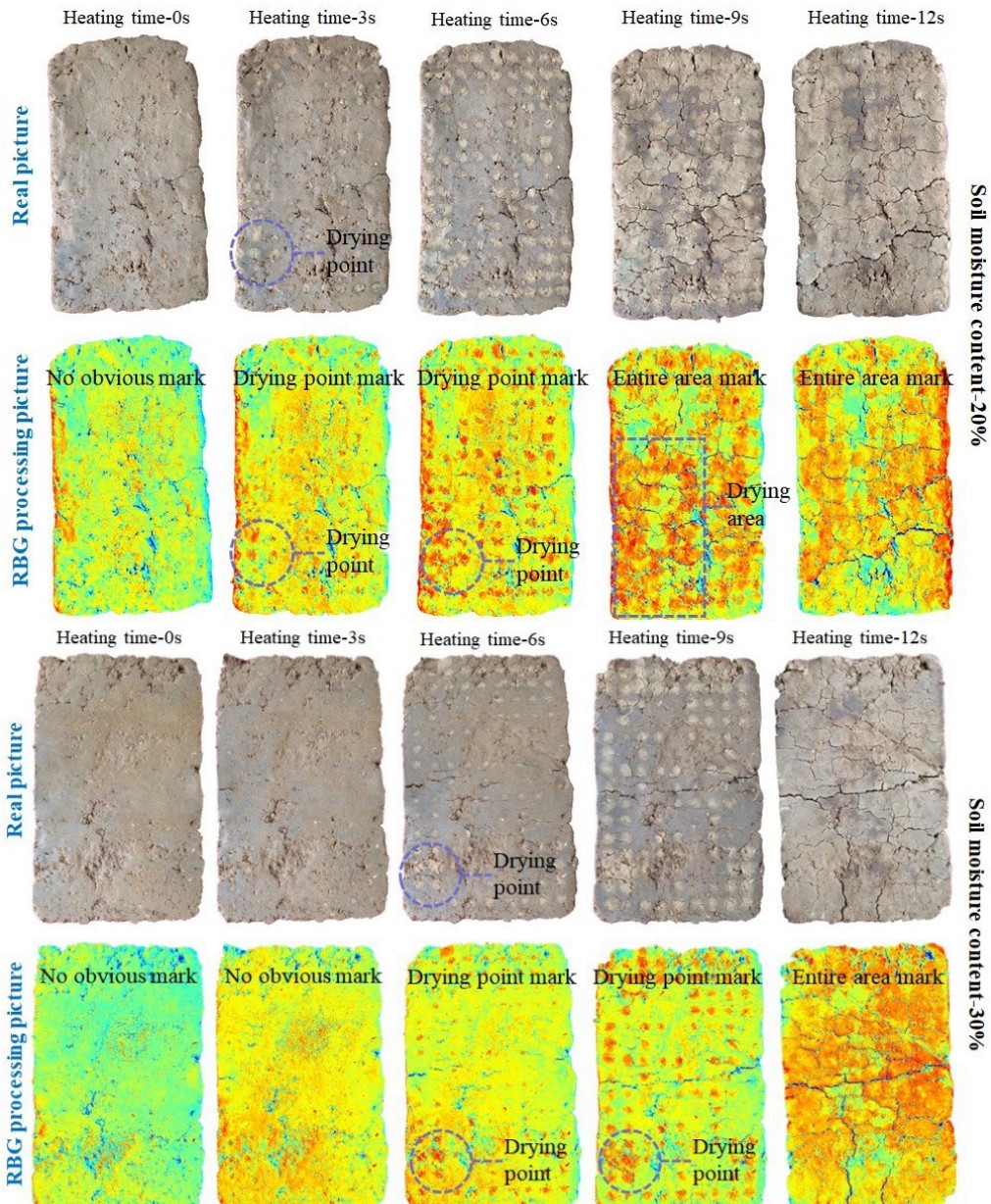

**Figure 14.** Changes in the surface of the subsoils with moisture contents of 20% and 30% when heated at 100 °C.

**Table 5.** Correlation of external friction with subsoil temperature, subsoil moisture content, and temperature gradient.

|  | Soil Moisture Content | 20% | 30% | 40% | Numerical Average |
|---|---|---|---|---|---|
| | Subsoil temperature and external friction angle | −0.918 | −0.745 | 0.990 | 0.884 |
| Correlation coefficient ($r_2$) | Temperature gradient and external friction angle | −0.915 | −0.746 | 0.879 | 0.847 |
| | Subsoil moisture content and external friction angle | 0.953 | 0.849 | −0.760 | 0.854 |

Overall, the above test results showed that the airflow temperature was positively correlated with the subsoil temperature and the subsoil temperature was negatively correlated with the subsoil moisture content. The higher the airflow temperature, the greater the reduction in the subsoil moisture content, and the greater the variations in the external friction angle and adhesive resistance between the tillage tool and the soil surface.

The moisture content of the soil is the key factor that determines whether the external friction angle increases or decreases after heating [44]. The external friction angle was the largest when the moisture content of sandy clay loam was approximately 30%. This result shows that the adhesion force between sandy clay loam and the subsoiling shovel was the largest under this condition. With 30% as the cut-off moisture content, any subsequent increase or decrease in the soil moisture content will decrease the external friction angle [45]. For the soil samples with a moisture content of 40%, the external friction angle increased after the samples were heated by hot air at a high temperature, even though the hot air dried out the subsoil surface. This is because the moisture content of the subsoil surface, although reduced under heating, was still greater than 30%. When the moisture content of the soil sample was 30%, the external friction angle reached the maximum value. After heating, the soil samples with 35% and 40% moisture content were dried and the moisture content tended to be 30%, so the external friction angle increased.

If the high-pressure hot air further reduces the moisture content of the subsoil to less than 30% during the operation of the tillage tool, the external friction angle between the tillage tool and soil will decrease, thereby reducing the working resistance [46]. However, if the soil moisture content is higher than 30%, and the subsoil moisture content is always greater than 30% after heating, it is recommended to apply only high-pressure airflow without heating to reduce resistance.

### 3.5. Results of the Soil-Bin Test on the Coupling Effect of High Pressure and Hot Air

The soil-bin test was conducted using a subsoiling shovel (in-soil depth of 150 mm). High-pressure (0.4 MPa) hot airflow (100 °C) was applied; the test results are shown in Figure 15. The average resistance to the shovel was 1031 N under the high-pressure hot air condition, the resistance was 1101 N with high-pressure airflow alone, and the resistance was 1210 N for the control group, for which the subsoiling shovel was used under normal conditions. Compared with the control group, the working resistance was reduced by 9.0% and 14.8% after the application of high-pressure airflow and the introduction of high-pressure hot air, respectively.

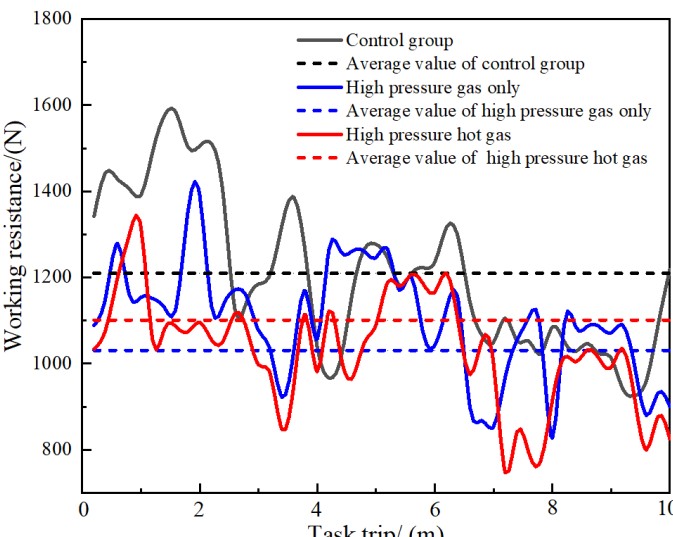

**Figure 15.** Working resistance tests on the subsoiling shovel under high-pressure hot air curtain and standard conditions.

The results show that the high-pressure hot airflow between the shovel and soil can reduce the working resistance. The coupling of high pressure and hot air has a significant resistance reduction effect. The high-pressure hot air curtain formed by the high-pressure hot airflow acts as a lubricant between the soil and subsoiling shovel [47]. Not only does it reduce the frictional resistance between the soil and shovel [48,49], but it also reduces the moisture content of the soil [50–52], thus reducing the adhesive resistance between the soil and shovel.

In addition, the working resistance was reduced by 9.0% after the high-pressure airflow was applied. On this basis, a further reduction of 5.8% can be achieved by heating the high-pressure airflow. This implies that for sandy clay loam, it is more effective to use a high-pressure air curtain as a lubricant in reducing resistance than heating [53,54].

High-pressure hot air can be used to reduce the resistance of the tillage tool, but it may also affect the quality of work. Therefore, further research will be conducted to study the coupling effect of high-pressure hot air on the operating quality of tillage tools.

## 4. Conclusions

In this study, high-pressure airflow was filled between a tillage tool and sandy clay loam to form a high-pressure air curtain layer, which acts as a lubricant that can reduce the external friction angle between the soil and tillage tool, thereby reducing the frictional resistance. The factors influencing the reduction in the external friction angle by the air curtain layer include the width of the air curtain layer, area of the air curtain layer, and roughness of the soil surface, among which the area of the air curtain layer is the most important. The greater the airflow pressure and the larger the air curtain layer area, the smaller the external friction angle and the greater the reduction in resistance.

A high-pressure hot air curtain layer was formed between the tool and soil owing to the coupling of heating with the high-pressure airflow. The hot air curtain increased the temperature at the subsoil surface, which reduced the moisture content of the subsoil in contact with the tillage parts. This caused changes in the adhesive resistance between the soil and tillage tool, thus affecting the external friction angle and working resistance. The variation pattern of the external friction angle is directly affected by the degree to which the moisture content of the subsoil is reduced. The cut-off moisture content for sandy clay loam is 30%. If the high-pressure hot air further reduces the moisture content of the subsoil to less than 30%, the external friction angle between the tillage tool and soil would be reduced. Conversely, if the subsoil moisture content cannot be reduced below 30%, the application of heating would conversely increase the external friction angle between the tillage tool and soil. Therefore, when using the high-pressure hot air method to reduce the working resistance of tillage parts, the air temperature parameters should be dynamically adjusted according to the soil moisture content to achieve the optimal drag reduction effect.

For the sandy clay loam with a moisture content of less than 30%, the high-pressure hot air achieved a great coupling effect; the working resistance of the subsoiling shovel was reduced by 14.8%. Test results have demonstrated that this high-pressure hot air approach can be used to reduce the resistance of tillage tools. Furthermore, the greatest effect on resistance reduction was achieved using the air curtain as a lubricant rather than by heating, which reduced resistance by reducing adhesion. The results of this study provided a theoretical and experimental basis for the development of tillage equipment with a high-pressure hot air resistance reduction function in sandy clay loam.

This research focused on analyzing the drag reduction mechanism of the high-pressure hot air approach but did not study the overall power consumption. Further research will explore the overall operation power consumption of this drag reduction method and further reduce the machine power consumption on the basis of ensuring the drag reduction effect, so as to achieve the effect of energy conservation and environmental protection.

**Author Contributions:** Conceptualization, K.Q.; methodology, K.Q., Y.Z., Z.S. and C.C.; software, Z.W.; validation, J.G., L.F. and H.B.; formal analysis, Y.Z., Z.S., C.C. and J.G.; investigation, data curation, K.Q., Y.Z. and Z.S.; writing—original draft preparation, K.Q. and Y.Z.; writing—review and editing, Z.S. and C.C.; visualization, J.G., L.F. and H.B.; supervision, K.Q., Z.S., C.C. and Z.W.; project administration, Y.Z., K.Q. and H.B.; funding acquisition, K.Q. All authors have read and agreed to the published version of the manuscript.

**Funding:** This work is supported by the National Natural Science Foundation of China [grant number 52105239]; the Open Fund of State Key Laboratory of Tea Plant Biology and Utilization [grant number SKLTOF20210121].

**Data Availability Statement:** Not applicable.

**Acknowledgments:** The authors are thankful to Zhao Wang for providing the test platform adjustment.

**Conflicts of Interest:** The authors declare no conflict of interest.

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
