# Peer review of "Investigating the Coupling Effect of High Pressure and Hot Air on External Friction Angle Based on Resistance Reduction Tests on Subsoiling Tillage Tools for Sandy Clay Loam"

_agronomy, doi:10.3390/agronomy12112663_

Round 1

Reviewer 1 Report

The paper is alright in the background. The authors are presenting the present a method of reducing friction for agricultural tools. The entire research plan presented in the article was well planned and implemented. However, the obtained results could be predicted without performing the presented tests. The presented research was aimed at finding and confirming the correctness of the use of air blowing on the blade during field work. In my opinion, the article lacks an analysis of the energy consumption during the movement of the blade in the soil with and without blowing hot air. The analysis should take into account the energy used to prepare a sufficient amount of compressed air at a given temperature. The obtained results of the spent energy for different working conditions of the blade will prove whether the applied method has a practical value and can be further developed.

Author Response

Dear reviewer:

Thank you for your decision and constructive comments on my manuscript. We have carefully considered the suggestion of Reviewer and make some changes. We have tried our best to improve and made some changes in the manuscript.

The red part that has been revised according to your comments. Revision notes, point-to-point, are given as follows:

Comments:

The paper is alright in the background. The authors are presenting the present a method of reducing friction for agricultural tools. The entire research plan presented in the article was well planned and implemented. However, the obtained results could be predicted without performing the presented tests. The presented research was aimed at finding and confirming the correctness of the use of air blowing on the blade during field work. In my opinion, the article lacks an analysis of the energy consumption during the movement of the blade in the soil with and without blowing hot air. The analysis should take into account the energy used to prepare a sufficient amount of compressed air at a given temperature. The obtained results of the spent energy for different working conditions of the blade will prove whether the applied method has a practical value and can be further developed.

Response:

Thank you for your comments. This research focused on analyzing the drag reduction mechanism of the high-pressure hot air approach, but did not study the overall power consumption. Further research will explore the overall operation power consumption of this drag reduction method, and further reduce the machine power consumption on the basis of ensuring the drag reduction effect, so as to achieve the effect of energy conservation and environmental protection. The relevant content has been added in the conclusion.(From line 509 to line 514)

Reviewer 2 Report

Thank the editor for giving me the opportunity to review the manuscript, so that I can learn from other scholars' research. The author studied the sandy clay loam, reduced the external friction angle of the soil by means of high-pressure hot air coupling, summarized the change rule of the external friction angle of soil samples with different moisture content, analyzed the images of the air curtain layer, studied the reasons for the reduction of the sliding friction angle, analyzed the reasons for the change of the external friction angle of the soil by means of the thermal image, and finally concluded that the high-pressure hot air coupling can reduce the subsoiling resistance. This paper is very well written, but there are also some problems.

1.     Before introducing the drag reduction method of high-pressure hot gas, the author needs to investigate whether this method will damage the soil environment. Relevant literature can be cited to explain in the introduction.

2.     When soil is exposed to air, its moisture content will change rapidly under the influence of many factors. How does the author maintain an ideal moisture content in soil samples? Please add in Section 2.2.

3.     The sliding friction is directly related to the pressure, that is, it is positively related to the quality of the soil sample. How can the author control this variable during the test?

4.     Please explain briefly why the sliding friction angle will increase when the moisture content of soil sample is higher than 30 and high pressure hot air is applied.

5.     It can be seen from Figure 14 that the heat transfer speed in the soil is slow during heating. How can the author ensure the effect of soil heating at the contact surface during the actual subsoiling operation?

6.     What method does the author use to classify soil particles?

7.     Some references are not standard, please check carefully.

In a word, this study is very meaningful, this manuscript can be accepted after minor revision.

Author Response

Dear reviewer:

We gratefully thanks for the precious time the reviewer spent making constructive remarks. We have tried our best to improve and made some changes in the manuscript. The red part that has been revised according to your comments. Revision notes, point-to-point, are given as follows:

  1. Before introducing the drag reduction method of high-pressure hot gas, the author needs to investigate whether this method will damage the soil environment. Relevant literature can be cited to explain in the introduction.

Response:

Thank you for your comments. After carefully studying your opinions and consulting relevant literature, we found that: Due to the abuse of chemical substances such as pesticides, the area of organic contaminated soil has increased year by year. Applying high temperature air to the soil can promote the volatilization of organic substances, improve the soil permeability, and help restore the contaminated soil. The relevant content has been added in the introduction. (From line 83 to line 86)

  1. When soil is exposed to air, its moisture content will change rapidly under the influence of many factors. How does the author maintain an ideal moisture content in soil samples? Please add in Section 2.2.

Response:

During the test, the moisture content of soil samples shall be measured regularly with the moisture meter, and the samples shall be sprayed with water regularly to ensure the stability of their moisture content. Related content has been added in section 2.2. (From line137 to line 139)

  1. The sliding friction is directly related to the pressure, that is, it is positively related to the quality of the soil sample. How can the author control this variable during the test?

Response:

Weighing the soil used to make the samples to ensure that each soil sample was of the same quality. Related content has been added in section 2.2.( (From line 130 to line 131)

  1. Please explain briefly why the sliding friction angle will increase when the moisture content of soil sample is higher than 30 and high pressure hot air is applied.

Response:

According to the previous study, it is known that the external friction angle has a great value when the moisture content of soil samples was 30%; after heating, soil samples with 35% and 40% moisture content were dried and the moisture content tends to 30%, so the external friction angle increases. The relevant content has been added at the end of section 3.4. (From line 442 to line 445)

  1. It can be seen from Figure 14 that the heat transfer speed in the soil is slow during heating. How can the author ensure the effect of soil heating at the contact surface during the actual subsoiling operation?

Response:

The small area of the deep pine shovel tip and the strong thermal conductivity of the metal can ensure that the airflow temperature on the shovel surface is high enough. The subsoil shovel keeps moving forward and the high pressure hot air keeps in close contact with the soil, so it can ensure that the deep pine shovel heats the soil while shearing the soil.

  1. What method does the author use to classify soil particles ?

Response:

Classification according to international soil classification standards. The relevant content has been added in section 2.2. (From line 129 to line 130)

  1. Some references are not standard, please check carefully.

Response:

Revisions have been made in accordance with journal requirements.

Reviewer 3 Report

It is a great honor to review this manuscript. This manuscript studied a new drag reduction method, which filled the high pressure hot air between the soil and the subsoiling shovel to reduce the external friction angle of the soil, and extracted the data of the air curtain layer through the image for further research. This research is very meaningful and provides a new idea for the future drag reduction.

Before this manuscript is accepted, there are still some mistakes that need to be corrected:

1. High pressure hot gas drag reduction is a very novel drag reduction method. Is there any other effect when high temperature gas plays a drag reduction role? Please add relevant research in the introduction.

2. In the actual test, the surface of the soil sample is not completely flat, and the uneven area on the sample will greatly affect the test results. What method does the author use to solve this problem?

3. Can you explain why the external friction angle is the largest when the soil moisture content is 30%? This is a very interesting phenomenon and is worth studying.

4. When measuring the external friction angle, vibration may occur when changing the angle, which will have a certain impact on the sliding friction angle. How does the author solve this problem?

5. The author's research on gas curtain layer is very detailed. Please briefly introduce what software the author used to obtain the data of gas curtain layer.

6. Some references are not relevant enough, and the author is suggested to change, for example (Tanner 1983 in line 437, Harsono, 2011 in line 445)

7. Also, there are few explanations of the rationale for the study design.

  This manuscript can be accepted after minor revision.

Author Response

Dear reviewer:

We gratefully appreciate for your valuable comments. We have tried our best to improve and made some changes in the manuscript. The red part that has been revised according to your comments. Revision notes, point-to-point, are given as follows:

  1. High pressure hot gas drag reduction is a very novel drag reduction method. Is there any other effect when high temperature gas plays a drag reduction role? Please add relevant research in the introduction.

Response:

According to existing research, the area of contaminated soil has increased due to the misuse of pesticides and other chemicals, and the application of hot air to the soil can promote the volatilization of organic matter and increase the infiltration rate of the soil, which can help restore the contaminated soil. The relevant content has been added in the introduction. (From line 83 to line 86)

  1. In the actual test, the surface of the soil sample is not completely flat, and the uneven area on the sample will greatly affect the test results. What method does the author use to solve this problem?

Response:

In the preparation of soil samples, the mold is placed on a smooth plane and pressed using a smooth pressing plate to ensure as smooth a soil sample as possible, and averaged over several trials during the test to minimize the influence of other factors.

  1. Can you explain why the external friction angle is the largest when the soil moisture content is 30%? This is a very interesting phenomenon and is worth studying.

Response:

According to the study, the adhesion force is inversely proportional to the growth of soil water content, and the adhesion force tends to be the largest when the soil water content is on the plastic limit (w=30%), and gradually decreases as the soil water content gradually increases. The relevant content has been added at the end of section 3.3. (From line 443 to line 445)

  1. When measuring the external friction angle, vibration may occur when changing the angle, which will have a certain impact on the sliding friction angle. How does the author solve this problem?

Response:

Pre-tests are conducted several times, and the angle is adjusted in advance according to the pre-test results, so that the vibration of the device can be minimized, and multiple tests are conducted to find the average value to reduce the error. Relevant contents have been added in Section 2.3. (From line 142 to line 146)

  1. The author's research on gas curtain layer is very detailed. Please briefly introduce what software the author used to obtain the data of gas curtain layer.

Response:

The background removal operation, contrast processing, image typology processing, and median filtering are performed in MATLAB, and the data such as the area width and roughness of the air curtain layer are obtained. Relevant contents have been added in Section 3.2. (From line 277 to line 278)

  1. Some references are not relevant enough, and the author is suggested to change, for example (Tanner 1983 in line 437, Harsono, 2011 in line 445)

Response:

We have selected more relevant literature for replacement according to your suggestions, namely [45] [46].

  1. Also, there are few explanations of the rationale for the study design.

Response:

This study investigated the coupling effect of high pressure and hot air on the ex-ternal friction angle by using a self-developed device to measure the external friction angle. Finally, the approach with the high-pressure hot air curtain was verified in tests on a subsoiling shovel. Before the formal test, the pre-test is conducted several times, and the average value of several tests is sought in the formal test to reduce the error. Relevant contents have been added in Section 2.3. (From line 142 to line 146)

Reviewer 4 Report

Quality article with good statistical processing of data.

I consider the manuscript to be of quite good quality, so I had no major comments.

Recommendation:

Table 2, Fig. 6 and others - Moisture content: it is advisable to add information (%of volume) in contrast to %by weight.

Units of the SI system: it is recommended to use m, mm instead of cm.

In the abstract, in the introduction and in the conclusions, it is appropriate to emphasize the novelty of the research solution.

It is advisable to adjust the form of citation of literary sources and references according to the guidelines for authors of publications in MDPI.

Author Response

Dear Reviewer:

Thank you for your recognition and encouragement of this paper, we are encouraged by your approval ! We have revised the manuscript according to your suggestions, and the relevant contents have been marked in the text.

1.Table 2, Fig. 6 and others - Moisture content: it is advisable to add information (%of volume) in contrast to %by weight.

Response:

  1. Thank you for your suggestion. Using mass water content as the object of study, it is easier to divide the interval and can ensure that most of the data are integers, which is convenient for statistical analysis. At this point if the mass water content is changed to volume water content, it will make a lot of data become decimal and difficult to observe the pattern. And volume water content and mass water content have a standard conversion formula, Readers can convert by themselves.
  2. Units of the SI system: it is recommended to use m, mm instead of cm.

Response:

  1. The unit has been replaced according to your suggestion, thank you again.
  2. In the abstract, in the introduction and in the conclusions, it is appropriate to emphasize the novelty of the research solution.

Response:

  1. Thank you for your suggestions,relevant content has been added in the abstract, introduction and conclusion.
  2. It is advisable to adjust the form of citation of literary sources and references according to the guidelines for authors of publications in MDPI.

Response:

  1. The references have been revised according to the requirements of the journal.

Reviewer 5 Report

The paper is very well written, this study investigated the coupling effect of high pressure and hot air on the external friction angle by using a self-developed device to measure the external friction angle. Test results showed that high-pressure air filled between the soil and tillage tool and formed a high-pressure air curtain layer, which acted as a lubricant, thereby reducing the external friction angle. The results of this study can provide both theoretical and experimental references for the application of the high-pressure hot air curtain in resistance reduction for tillage tools.

There are some problems, which must be solved before it is considered for publication.

1. Why does the author set the water content of the soil tank test bench to 25% when conducting deep loosening test?

2. In Section 3.4, the author observed the images of the hot air curtain formed by the high-pressure airflow at different temperatures, but according to the thermal imaging images, this should be the thermal imaging image of the soil sample after heating. Please describe it accurately.

3. There are a large number of microorganisms living in the soil. Will high pressure hot gas damage these microorganisms, and then damage the soil?

4. When using the external friction angle measuring device, once the air curtain layer is generated, the external friction angle of the soil will change rapidly. How does the author adjust the angle of the device in time?

5. The research object of this paper is sandy clay loam. How to keep the characteristics of sandy clay loam when making samples?

6. According to the author's description, the so-called air curtain area should be the area of the cross section of the air curtain taken by the micro camera. It is suggested that the author standardize his statement.

7. Can you simply explain the reason why the sliding friction angle changes after heating when the moisture content of the soil sample is higher than 30%? This unconventional phenomenon is very interesting. The author can make it simple for readers to understand.

8. What are the further research plans for the authors of this study?

9. The format of the following references should be standardized according to the requirements of the journal.

Author Response

Dear Reviewer:

Thank you for your valuable comments. We have revised the paper according to your comments. The red part that has been revised according to your comments. Revision notes, point-to-point, are given as follows:

  1. Why does the author set the water content of the soil tank test bench to 25% when conducting deep loosening test?

Response:

At the time of soil extraction, the soil moisture content was measured to be 25%, and to maintain the authenticity of the soil tank test results, the soil water content in the tank was also set to 25%. Relevant contents have been added in Section 2.3.3.(line231)

  1. In Section 3.4, the author observed the images of the hot air curtain formed by the high-pressure airflow at different temperatures, but according to the thermal imaging images, this should be the thermal imaging image of the soil sample after heating. Please describe it accurately.

Response:

The image is a thermal image of the soil sample, not of the air curtain layer, and has been modified in the manuscript in response to comments. Relevant contents have been added in Section 3.4. (line 361)

  1. There are a large number of microorganisms living in the soil. Will high pressure hot gas damage these microorganisms, and then damage the soil?

Response:

Few of the current tillage methods operate with full coverage of agricultural soil, so the impact on the range of microbial activity is small. According to existing research, the area of contaminated soil has increased due to the misuse of pesticides and other chemicals, and the application of hot air to the soil can promote the volatilization of organic matter and increase the infiltration rate of the soil, which can help restore the contaminated soil. The relevant content has been added in the introduction. (From line 83 to line 86)

  1. When using the external friction angle measuring device, once the air curtain layer is generated, the external friction angle of the soil will change rapidly. How does the author adjust the angle of the device in time?

Response:

Pre-tests were conducted several times, and the angle are adjusted in advance according to the pre-test results, and then fine-tuned during the actual test, and averaged over several tests to reduce errors. Relevant contents have been added in Section 2.3. (From line 145 to line 146)

  1. The research object of this paper is sandy clay loam. How to keep the characteristics of sandy clay loam when making samples?

Response:

Based on the proportion of clay, silt and sand content in the soil, these three soil particles are prepared using a standard soil sieve and then mixed and stirred to maximize the restoration of soil properties.

  1. According to the author's description, the so-called air curtain area should be the area of the cross section of the air curtain taken by the micro camera. It is suggested that the author standardize his statement.

Response:

The area of the front view of the air curtain layer has been modified according to the suggestion. Relevant contents have been added in Section 3.2. (line 278)

  1. Can you simply explain the reason why the sliding friction angle changes after heating when the moisture content of the soil sample is higher than 30%? This unconventional phenomenon is very interesting. The author can make it simple for readers to understand.

Response:

According to the previous study, it is known that the external friction angle has a great value when the moisture content of soil samples was 30%; after heating, soil samples with 35% and 40% moisture content were dried and the moisture content tends to 30%, so the external friction angle increases. Relevant contents have been added in Section 3.4. (From line 442 to line 445)

  1. What are the further research plans for the authors of this study?

Response:

Further research will explore the overall operation power consumption of this drag reduction method, and further reduce the machine power consumption on the basis of ensuring the drag reduction effect, so as to achieve the effect of energy conservation and environmental protection. (From line 509 to line 514)

  1. The format of the following references should be standardized according to the requirements of the journal.

Response:

Revisions have been made in accordance with journal requirements.
